# Shockwaves Suppress Adipocyte Differentiation via Decrease in PPARγ

**DOI:** 10.3390/cells9010166

**Published:** 2020-01-09

**Authors:** Wonkyoung Cho, SeoYeon Kim, Myeongsook Jeong, Young Mi Park

**Affiliations:** Department of Molecular Medicine, College of Medicine, Ewha Womans University, Seoul 03760, Korea; cwkdyb@gmail.com (W.C.); aodtnr1011@naver.com (M.J.)

**Keywords:** adipocyte differentiation, shockwaves, PPARγ, 3T3L-1, cAMP

## Abstract

Adipogenesis is a crucial cellular process that contributes to the expansion of adipose tissue in obesity. Shockwaves are mechanical stimuli that transmit signals to cause biological responses. The purpose of this study is to evaluate the effects of shockwaves on adipogenesis. We treated 3T3L-1 cells and human primary preadipocytes for differentiation with or without shockwaves. Western blots and quantitative real-time reverse transcriptase PCR (qRT-PCR) for adipocyte markers including peroxisome proliferator-activated receptor γ (PPARγ) and CCAAT-enhancer-binding proteins (C/EBPα) were performed. Extracellular adenosine triphosphate (ATP) and intracellular cyclic adenosine monophosphate (cAMP) levels, which are known to affect adipocyte differentiation, were measured. Shockwave treatment decreased intracellular lipid droplet accumulation in primary human preadipocytes and 3T3-L1 cells after 11–12 days of differentiation. Levels of key adipogenic transcriptional factors PPARγ and/or C/EBPα were lower in shockwave-treated human primary preadipocytes and 3T3L-1 cells after 12–13 days of differentiation than in shockwave-untreated cells. Shockwave treatment induced release of extracellular ATP from preadipocytes and decreased intracellular cAMP levels. Shockwave-treated preadipocytes showed a higher level of β-catenin and less PPARγ expression than shockwave-untreated cells. Supplementation with 8-bromo-cAMP analog after shockwave treatment rescued adipocyte differentiation by preventing the effect of shockwaves on β-catenin, Wnt10b mRNA, and PPARγ expression. Low-energy shockwaves suppressed adipocyte differentiation by decreasing PPARγ. Our study suggests an insight into potential uses of shockwave-treatment for obesity.

## 1. Introduction

Adipose tissue regulates energy and metabolic homeostasis in the body. White adipose tissue is a fat-storage organ for energy and an endocrine organ for secretion of adipokines including leptin, adiponectin, resistin, and other factors [1]. Adipose tissue is composed of adipocytes, adipose stem cells, and other cell types including vascular smooth muscle cells, endothelial and neuronal cells [2]. Among these cells, adipocytes generated through adipogenesis are the main component of adipose tissue. Adipogenesis is the process of commitment of mesenchymal stem cells (MSCs) to preadipocytes and achievement of terminal differentiation through which preadipocytes gain characteristics of mature adipocytes [3].

Preadipocyte differentiation into adipocytes is regulated by transcriptional cascades. Molecules for cell shape and extracellular matrix remodeling also contribute to differentiation [1]. The major adipogenic transcriptional factors are peroxisome proliferator-activated receptor γ (PPARγ) and CCAAT-enhancer-binding proteins (C/EBPs), which induce mature adipogenic phenotypes such as lipid storage and synthesis, lipolysis, and adipokine secretion [4,5,6]. Adipogenesis is a complicated process regulated by transcriptional activators and repressors for PPARγ and C/EBPα expression [7,8].

Derived from mouse embryos, 3T3L-1 preadipocytes are widely used to study adipocyte differentiation because of their immortality and key features of in vivo mature adipocytes [9]. After growth arrest at confluence, preadipocyte differentiation is initiated by adding insulin, dexamethasone (DEXA), and 3-isobutyl-1-methylxanthine (IBMX). Insulin and DEXA are stimulators for insulin-like growth factor 1 (IGF-1) and glucocorticoid receptor. IBMX increases intracellular cyclic adenosine monophosphate (cAMP) [9].

Wnt ligands, the secreted glycoproteins are extracellular signaling molecules that inhibit adipogenesis. Wnt binding to receptors induces phosphorylation and inactivation of glycogen synthase kinase 3β (GSK3β) [10]. Wnts signal through the frizzled receptor and low-density lipoprotein receptor-related protein co-receptors [11,12]. Wnt signaling stabilizes cytoplasmic β-catenin and induces translocation of accumulated β-catenin into the nucleus. Beta-catenin binding to T-cell factor/lymphoid-enhancing factor induces expression of downstream target genes. In the absence of Wnt, degradation complexes containing dephosphorylated GSK3β inactivate β-catenin by phosphorylation and lead to proteasomal degradation of β-catenin [13,14,15].

Shockwaves are a sequence of mechanical pulses characterized by high peak pressure (100 MPa), fast rise (<10 ns), and short lifecycle (10 µs) [16]. High energy shockwaves were developed for lithotripsy. Later on, low-energy shockwave therapy has been proven to be beneficial for several medical conditions including orthopedic diseases [17]. In orthotripsy for musculoskeletal diseases, shockwaves induce extracellular biological responses and tissue regeneration. Recent studies revealed that the therapeutic effects of shockwave treatment are derived from neovascularization, differentiation of mesenchymal stem cells, and releases of angiogenetic factors [17,18]. Extracorporeal shockwave treatment (ESWT) promotes tissue repair through modulating a variety of gene expressions. Shockwaves decrease matrix metalloproteinases and interleukins and produces a regenerative effect in musculoskeletal tissues [17]. In tendinitis repair, shockwaves increase transforming growth factor beta 1 (TGFβ1) and insulin-like growth factor 1 (IGF-1) [19]. Low energy shockwaves increase expressions of proliferating cell nuclear antigen (PCNA), collagen type I, collagen type II, and TGFβ1 and release nitric oxide (NO) and TGFβ [20]. Shockwave treatment also promotes osteogenesis through adenosine triphosphate (ATP) release and inhibits chondrogenic differentiation through adenosine release and activation of A2B receptor in human mesenchymal stem cells (hMSCs) [21,22].

Obesity is a serious medical condition in which body fat mass is increased. It is related to metabolic diseases such as type II diabetes, cardiovascular diseases, and cancer [23]. Obesity is characterized by excessive accumulation of white adipose tissue (WAT). Adipose tissue growth is resulted from hypertrophy and hyperplasia of adipocytes. Many mouse models of obesity proved that both the adipocyte sizes and numbers are increased in obese adipose tissue [24]. While obesity progresses, Zucker rats show progressive adipocyte hypertrophy until the adipocytes reach crucial cell sizes and adipocyte hyperplasia is followed thereafter [25]. The in vitro model of human adipocytes proved that pre-adipocytes from elderly humans keep the ability of differentiation into adipocytes and contribute to adipose tissue growth [26]. Therefore, understanding the mechanism of adipogenesis is principal for developing a therapeutic strategy for the treatment of obesity.

In the current study, we tested the effect of shockwaves on adipocyte differentiation and found that low-energy shockwaves suppressed adipocyte differentiation by decreasing PPARγ and C/EBPα in 3T3L-1 and primary human preadipocytes.

## 2. Materials and Methods

### 2.1. Reagents

Oil-red-O, 8-Bromoadenosine 3’,5’-cyclic adenosine monophosphate (8-bromo-cAMP), suramin, insulin, DEXA and IBMX were purchased from Sigma (St. Louis, MO, USA).

### 2.2. Cell Culture and Differentiation (3T3L-1 Mouse Preadipocytes and Primary Human Subcutaneous Preadipocytes)

3T3L-1 mouse preadipocytes were purchased from the American Type Culture Collection (ATCC, CL-173) (Manassas, VA, USA) and cultured in Dulbecco’s Modified Eagle’s Medium (DMEM) supplemented with 10% (*v*/*v*) new born calf serum (NBCS) (Gibco, Waltham, MA, USA), 1% (*v*/*v*) penicillin and streptomycin (P/S). Cells were cultured at 37 °C, in a 5% CO_2_ incubator. At 70–80% confluency, preadipocytes were trypsinized, resuspended, and placed in culture plates for differentiation into adipocytes. In a 6-well plate, 1 × 10^5^ preadipocytes were seeded per well and on the next day we washed the cells with pre-warmed Dulbecco’s phosphate buffered saline (DPBS) and changed the medium to DMEM containing 10% (*v*/*v*) NBCS and 1% (*v*/*v*) P/S. For a 12-well plate, 0.075 × 10^5^ preadipocytes per well were seeded. Two days after the cell seeding, the cells became confluent. After 2 days post-confluent, we added insulin (1 µg/mL), DEXA (1 µM) and IBMX (0.5 mM) to DMEM containing 10% (*v*/*v*) fetal bovine serum (FBS) and 1% P/S (*v*/*v*) (Gibco) to induce differentiation of the 3T3L-1 preadipoctyes into adipocytes. After 2 days, medium was replaced with adipocyte maintenance medium which is DMEM containing 10% FBS and 1% (*v*/*v*) P/S. The medium was replenished with DMEM (10% FBS and 1% P/S) every 2–3 days.

Primary human subcutaneous preadipocytes (ATCC PCS-210-010^TM^) were from ATCC (Manassas, VA, USA). We followed the supplier’s instructions. Briefly, when the human preadipocytes reached 70–80% confluence, cells were trypsinized and seeded into a culture plate at a density of 18,000 cells/cm^2^. For differentiation, an ATCC toolkit (Cat# ATCC PCS-500-050) was used. Two days after the seeding, adipocyte differentiation initiation medium from ATCC was added into each well. After 2 days, the medium was replaced with adipocyte differentiation maintenance medium from ATCC. Every 3–4 days, medium was replenished with maintenance medium until the cells reached full differentiation.

### 2.3. Shockwave Treatment

A Dornier AR2 ESWT (Dornier MedTech, Weßling, Germany) device was used for shockwave treatment. Each well of a culture plate with confluent 3T3L-1 cells was fully filled with media and shockwaves were applied through the contact between the head of the shockwave device and the surface of the medium. After shockwave treatment, a proper volume of media was left for cell culture.

For Annexin V staining, Adenosine triphosphate (ATP) measurement and cAMP measurement, and MTT assay, 3T3-L1 preadipocytes were cultured and treated with shockwaves like below; 3T3L-1 preadipocytes were seeded at a density of 1 × 10^5^ cells per well in a 6-well culture plate. On the next day, the medium was replenished after washing with pre-warmed DPBS. Two days after the cell seeding, the cells got confluent. After 2 days post-confluent, the cells were treated with or without shockwaves (0.025 mJ/mm^2^, 10 Hz, 1000 impulses).

### 2.4. Annexin V Staining

3T3L-1 cells were trypsinized and stained with fluorescein isothiocyanate (FITC) annexin V and PI (Propidium Iodide) one day after the shockwave treatment (FITC annexin V apoptosis detection kit; BD Biosciences, San Jose, CA, USA). Flow cytometry was performed by using a FACS Aria (BD Biosciences).

### 2.5. Adenosine Triphosphate (ATP) Measurement

Cells were treated with or without shockwaves as described in Section 2.3. Media was collected after the shockwave treatment. Suramin (200 μM), an ATPase inhibitor, was added into the media. Then, ATP was measured using ATP fluorometric assay kits (Abcam, Cambridge, UK). Fluorescence was measured with microplate reader (Biotek, Winooski, VT, USA).

### 2.6. cAMP Measurement

Shockwave-treated or untreated cells were collected and intracellular cAMP was measured using the cAMP assay kit (Abcam). A microplate reader was used (Biotek, Winooski, VT, USA).

### 2.7. Oil-Red-O Staining

Cells were fixed in 4% (*v*/*v*) paraformaldehyde in phosphate buffered saline (PBS) for 20 min at room temperature, washed with PBS, and stained with 0.5% (*w*/*v*) Oil-red-O solution for 20 min. After rinsing with distilled water, images were captured via light microscope. To measure the Oil-red-O incorporated into intracellular lipids, we added an equal volume of isopropanol to cells for 5 min and extracted Oil-red-O. Absorbance (optical density) of Oil-red-O in the extract was measured using spectrophotometry at 510 nm (Biotek). Oil-red-O values which represent intracellular lipid amounts were normalized to the DNA quantity of the cells.

### 2.8. MTT Assay

To count live cells at Day 12, we performed an MTT assay using fully differentiated adipocytes. O.D was measured at 540 nm by a microplate reader (Biotek).

### 2.9. RNA Isolation and Quantitative Real-Time PCR (qRT-PCR)

Total RNA was extracted from 3T3L-1 cells using TRIzol reagent according to the manufacturer’s instructions (Invitrogen, Waltham, MA, USA). We synthesized cDNA using an iScript cDNA synthesis Kit (Bio-Rad, Hercules, CA, USA). For qRT-PCR, power SYBR^®^ Green PCR Master Mix (Applied Biosystems, Foster City, CA, USA) and an ABI Real-Time PCR thermocycler were used. After qRT-PCR, results were normalized to gyceraldehyde-3-phosphate dehydrogenase (GAPDH). All qRT-PCR analyses were repeated at least 3–4 times. Primers were listed in the table below (Table 1).

### 2.10. Immunoblot Analyses

3T3L-1 cells were lysed in a buffer containing 20 mM Tris-HCl (pH 7.5), 150 mM NaCl, 1 mM ethylenediaminetetraacetic acid (EDTA), 1 mM ethylene glycol tetraacetic acid (EGTA), and 1% (*v*/*v*) Nonidet P-40 (NP-40). A protease inhibitor cocktail (Roche) and phosphatase inhibitors (10 mM phenylmethylsulfonyl fluoride, 1% sodium pyrophosphate, 10 mM sodium fluoride, and 2 mM sodium vanadate) were added. By SDS-polyacrylamide gel electrophoresis (PAGE), 20 µg proteins were separated, then transferred to PVDF membranes (Millipore, Burlington, MA, USA) for immunoblotting. Membranes were probed with antibodies for PPARγ (#2435), C/EBPα (#8178), perilipin (#9349), fatty acid synthase (FAS) (#3180), acetyl-CoA carboxylase (ACC) (#3676) and caspase 3 (#9662) (Cell Signaling, Danvers, MA, USA). Protein concentration was measured with Pierce™ BCA Protein Assay Kit (Thermo Fisher Scientific, Walthom, MA, USA). As control, β-actin (sc-47778) (Santa Cruz, CA, USA) was used for normalization. Band intensities were quantified using ImageJ (http://rsbweb.nih.gov/ij).

### 2.11. Statistical Analysis

Data are expressed as mean + standard deviation (S.D.). For statistical analysis, an unpaired two-tail Student *t*-test was used. *P*-values less than 0.05 were considered significant. All experiments were repeated at least 3–4 times. In all panels, levels of indicated protein or RNA were normalized to β-actin or GAPDH and expressed as fold-increase relative to a control that was arbitrarily set at 1 (100%). Analyses were performed using GraphPad Prism Software (version 5, GraphPad Software, San Diego, CA, USA).

## 3. Results

### 3.1. Shockwaves Suppressed Adipocyte Differentiation of Murine 3T3L-1 and Human Primary Preadipocytes

To determine if shockwaves affect adipocyte differentiation, 3T3L-1 and human primary preadipocytes were treated with or without shockwaves (0.025 mJ/mm^2^, 10 Hz, 1000 impulses) after inducing adipocyte differentiation and whenever the medium was changed. Differentiation into adipocytes was induced by adding insulin, DEXA, and IBMX (Figure 1A). Intracellular lipid droplet accumulation, a main phenotype of mature adipocytes, was visualized and evaluated by Oil-red-O staining, 11 days after induction of adipocyte differentiation (Figure 1B). Shockwave treatment reduced lipid accumulation in differentiated 3T3-L1 cells on day 11. Quantification of the optical density of Oil-red-O extracted by isopropanol revealed that shockwave-treated 3T3L-1 cells had 15% smaller amounts of lipids than untreated cells on day 11. Microscopic images (5×, 20×) showed that shockwave-treated 3T3-L1 cells are smaller in size having smaller lipid droplets in their cytoplasms compared to untreated 3T3-L1 cells. The numbers of differentiated 3T3-L1 cells were not significantly different between shockwave-treated and untreated 3T3-L1 cells (Appendix A). Primary human preadipocytes were also differentiated with or without shockwave-treatment. Oil-red-O staining and isopropanol extraction on day 12 showed 15% less lipid accumulation in shockwave-treated cells compared to untreated cells (Figure 1C).

### 3.2. Shockwaves Decreased PPARγ, A Key Adipogenic Transcriptional Factor, in 3T3-L1 and Human Preadipocytes

Preadipocyte differentiation is a complex process that involves coordinated expression of specific genes and proteins functioning at each stage of differentiation. To analyze the effect of shockwaves in adipocyte differentiation, we performed qRT-PCR for adipogenesis markers including key transcription factors (PPARγ 1/2 and C/EBPα), lipid droplet proteins (perlipin1 and Fsp27), lipid synthesis proteins (FAS, ACC), and a preadipocyte marker (PREF1). PPARγs have two isoforms: PPARγ1 and PPARγ2. PPARγ1 is expressed in most tissues, and PPARγ2 expression is limited to adipose tissue [27]. Shockwaves induced decreases in the adipocyte markers PPARγ1, PPARγ2, C/EBPα, perilipin1, Fsp27, FAS, and ACC at the RNA level (Figure 2A). The level of preadipocyte marker PREF1 was 1.7-fold higher in shockwave-treated 3T3L-1 cells than in untreated cells. We also measured adipocyte markers by Western blot and found that PPARγ1/2, C/EBPα, Perilipin, FAS, and ACC were less in shockwave-treated 3T3L-1 cells compared with shockwave-untreated cells consistent with the results in Figure 2A (Figure 2B).

Western blots using primary human preadipocytes showed 40% less PPARγ in shockwave-treated cells than in untreated cells on day 13 (Figure 2C). Overall, these data revealed that low energy shockwaves (0.025 mJ/mm^2^, 10 Hz, 1000 impulses) downregulated PPARγ and suppressed preadipocyte differentiation into adipocytes. However, when we treat fully differentiated 3T3-L1-derived adipocytes (day 14) with shockwaves, the shockwave treatment did not affect the level of PPARγ (Appendix A).

### 3.3. Shockwave Treatment at the Initial Stage of Adipocyte Differentiation Suppressed PPARγ Expression

To see time-course changes of PPARγ and C/EBPα expression during differentiation, shockwave-treated and untreated 3T3L-1 cells were harvested at the indicated timepoints (Figure 3A). Immunoblots showed that PPARγ and C/EBPα (28, 42 kDa) increased 2 days after initiation of differentiation. Shockwave-treated 3T3L-1 preadipocytes showed 50% less increase in PPARγ and 75% less increase in C/EBPα than untreated cells.

To determine when the shockwaves should be given for the most effective suppression of PPARγ, we tested different schedules of shockwave treatment during adipocyte differentiation (Figure 3B). The first three shockwave treatments (D0, D2, D4) induced the lowest PPARγ levels. Additional shockwave treatment after seven days of differentiation did not have an additive effect on PPARγ decrease. To confirm the efficiency of early shockwave treatment, shockwaves were given three times at early or late phases (Figure 3C). The 3T3L-1 cells treated with shockwaves three times during the early phase showed similar results as entirely six-time shockwave-treated cells. The last three times of shockwave treatment in the late phase did not have a PPARγ decreasing effect. We also tested different energy levels of shockwaves during adipocyte differentiation as in Figure 1A. Immunoblotting for PPARγ showed that shockewaves suppressed PPARγ expression in an energy-level dependent manner (Figure 3D).

### 3.4. Shockwave Treatment Did Not Affect Cell Viability

We tested if shockwaves, the mechanical pulses affect cellular viability. We stained shockwave-treated and untreated 3T3-L1 preadipocytes with Annexin V and found that low-energy shockwaves (0.025 mJ/mm^2^) did not cause apoptosis (Annexin V-positive cells) nor necrosis (PI-positive cells) (Figure 4A). We also counted viable cell number with an MTT assay, and the result showed no significant difference between shockwave-treated and untreated 3T3-L1 cells on day 12 (Figure 4B). Immunoblotting for caspase 3 using shockwave-treated and untreated 3T3-L1 cells on day 14 showed no activation of caspase 3 (Figure 4C). Therefore, we confirmed that the low-energy shockwaves did not induce cell death during the course of adipocyte differentiation.

### 3.5. Shockwaves Promoted Extracellular ATP Release from Preadipocytes and Decreased Intracellular cAMP in 3T3L-1 Cells

IGF-1 and glucocorticoids are factors that promote adipocyte differentiation. Wnts, TGFβ, and inflammatory cytokines inhibit differentiation. Shockwaves induce extracellular ATP release from various cell types, which is one way in which shockwaves induce cell signaling [21,28]. To evaluate if shockwaves induced ATP release in 3T3L-1 preadipocytes, extracellular ATP was measured at indicated timepoints after shockwave treatment (Figure 5A). Extracellular ATP concentration in the media increased significantly after the shockwave treatment.

Intracellular ATP is converted into cAMP by adenylyl cyclase. Intracellular cAMP is a second messenger for adipocyte differentiation and induces PPARγ transcription [29]. IBMX increases intracellular cAMP to promote preadipocyte differentiation into adipocytes [4]. We measured intracellular cAMP levels in shockwave-treated or untreated 3T3-L1 preadipocytes, 3 and 10 min after adding IBMX (0.5 mM) in the media which did not contain insulin and DEXA. Shockwave treatment suppressed the intracellular cAMP increase by IBMX (Figure 5B).

Extracellular ATP binds to purinergic receptors and results in diverse physiological effects [30]. A previous report showed that treating proliferating 3T3L-1 preadipocytes (less than 50% confluence) with extracellular ATP promotes adipocyte differentiation in response to adipogenic hormones [31]. In contrast, the conventional protocol that we used for adipocyte differentiation requires cell cycle arrest of preadipocytes (100% confluence) before differentiation. To investigate if extracellular ATP released by shockwave treatment affected the shockwave-induced inhibition of adipocyte differentiation, we added different concentrations of ATP into the media instead of shockwave treatment. Extracellular ATP did not affect PPARγ expression, while shockwaves reduced PPARγ expression of 3T3-L1 cells on day 10 of differentiation (Figure 6).

### 3.6. Shockwaves Activated Wnt Signaling through β-Catenin by Decreasing Intracellular cAMP

Wnts are signaling molecules that stabilize β-catenin and inhibit adipogenesis. Intracellular cAMP inhibits the Wnt signaling pathway by decreasing Wnt10b mRNA [5]. We examined if the decreased cAMP levels after shockwave treatment affected the Wnt pathway that regulates β-catenin. An endogenous activator of Wnt signaling is Wnt10b, which declines upon induction of adipocyte differentiation [32]. After 6 h of differentiation, Wnt10b mRNA in shockwave-untreated 3T3-L1 cells was decreased by more than 50%. Wnt10b mRNA levels of shockwave-treated 3T3-L1 cells did not decline and 80% of the baseline level was sustained (Figure 7A). To test if the effect of shockwaves on Wnt10b was through decreased cAMP, we added 8-bromo-cAMP, a cAMP analog to 3T3-L1 cells just after the shockwave treatment. The 8-bromo-cAMP supplement induced a decline in Wnt10b mRNA in shockwave-treated 3T3-L1 cells, suggesting that shockwave-induced cAMP decrease caused conservation of Wnt10b (Figure 7A).

To examine time-course changes in β-catenin, 3T3L-1 cells were harvested at the indicated timepoints during adipocyte differentiation. Immunoblotting showed 2- to 3-fold higher β-catenin levels in shockwave-treated 3T3L-1 cells on days 1 and 2 of differentiation compared to untreated cells (Figure 7B). PPARγ expression is induced four days after differentiation. Shockwave-treated 3T3L-1 cells showed about 40% less PPARγ on days 7 and 12 of differentiation than untreated 3T3L-1 cells (Figure 7C). The addition of 8-bromo-cAMP blocked the effect of the shockwaves on β-catenin. PPARγ expression in shockwave-treated 3T3-L1 cells was induced to levels comparable to shockwave-untreated control cells when cAMP was complemented (Figure 7B,C).

We concluded that a shockwave-induced decrease of cAMP inhibited preadipocyte differentiation into adipocytes via conservation of Wnt10b and freed function of β-catenin.

## 4. Discussion

Shockwaves are mechanical pulses characterized by extremely high amplitude with short rise time, followed by long, low-magnitude negative waves [16]. Extracorporeal shockwave treatment was introduced for lithotripsy in the 1980s [33,34]. While high-energy shockwaves are used for lithotripsy, low-energy shockwaves are reported to induce improvement of symptoms for clinical conditions including orthopedic and soft tissue diseases [35,36]. Effects of mechanical forces on cell fate and differentiation have been studied [37,38,39]. High frequency and very low-magnitude mechanical signals reduce adiposity in mice [40]. Mechanical strain increases β-catenin, which suppresses PPARγ in MSCs [37]. In addition, mechanical loading such as shear stress contributes to osteogenesis signaling pathways through Wnt, IGF-I, estrogen receptor (ER), and bone morphogenetic protein (BMP) [41]. Shockwaves induce osteogenesis of human MSCs [21,41].

Obesity is a major risk factor for metabolic diseases including cardiovascular disease and type 2 diabetes [42,43,44,45]. White adipose tissue (WAT) is a multifactorial organ that regulates various metabolic functions [46]. Physiological functions of WAT are impaired by inflammation, fibrosis, hypoxia, dyregulated adipkine secretion and lipotoxicity in obesity [42]. This induces insulin resistance and leads to development of type 2 diabetes.

Increasing fat mass is resulted from increased sizes and numbers of adipocytes. Adipogenesis is the process in which preadipocytes differentiate into mature adipocytes. The integrity of adipocytes is maintained by balance between adipogenesis of preadipocytes and apoptosis of adipocytes throughout life time. In animal studies, a white adipocyte number increases during puberty and the number of adipocytes is kept stable in adult adipose tissue [47]. In human, about 10% of adipocytes undergo annual turnover [48]. In animals, adipocyte sizes increase upon high fat diet and the increase of adipocyte number follows thereafter [49,50]. An increase in adipocyte number is also observed in human adipose tissues after short-term overfeeding [51]. Moreover, the analysis of WAT from obese individuals revealed that adipocyte size and number are highly correlated with the risk for metabolic syndrome, independent of body mass index (BMI) [52,53]. Nevertheless, adipogenesis seems to be a crucial component for pathologic obesity, and adipogenesis inhibition has been regarded as a strategy in the obesity treatment. There have been many studies for revealing mechanisms of adipogenesis and developing adipogenesis inhibitors [54]. However, physiological mechanisms regulating adipocyte number in adulthood are not clearly defined and anti-adipogenesis drugs with high effectiveness have not yet been developed.

Preclinical and human studies have shown that weight loss is related with decreased sizes of adipocytes; however, it is not related with adipocyte number [48,50]. While adipogenesis is certainly a part of pathologic WAT remodeling, increasing the number of adipocytes contributes to healthy adipose tissue expansion characterized by increased adipose storage capacity, observed in the metabolically healthy obesity [55]. Therefore, intense studies are warranted for revealing mechanisms of adipogenesis and roles of adipogenesis both in the healthy and pathologically obese states.

In the current study, we evaluated the effect of shockwaves in adipogenesis and found that low-energy shockwaves (0.025 mJ/mm^2^, 10 Hz, 1000 impulses) suppressed preadipocyte differentiation into adipocytes (Figure 1, Figure 2 and Figure 3). The low-energy shockwaves did not induce apoptosis or death of the cells in our experiment (Figure 4). Confluent 3T3L-1 preadipocytes initiate differentiation on exposure to insulin, DEXA, and IBMX. This differentiation cocktail induces two phases of differentiation. The first is clonal expansion with preadipocytes entering the cell cycle and expressing adipogenic factors to converge on master regulators PPARγ and C/EBPα. The second phase is terminal cell differentiation with an expression of mature adipocyte phenotypes including lipid droplets [1]. Adipogenesis is a complex and highly orchestrated program involving transcription factors. PPARγ-deficient embryonic stem cells are lethal due to a problem in placentation [56]. Hypomorphic PPARγ-null mice show severe lipodystrophy [57]. C/EBPα-deficient mice die soon after birth due to problems in glucose production. C/EBPα is required for gluconeogenesis in the liver [58]. We found that low-energy shockwaves prevented differentiation of murine 3T3L-1 and primary human preadipocytes by suppressing PPARγ and C/EBPα expression (Figure 1 and Figure 2). Decreased protein expression of ACC, FAS, and perilipin related to lipogenesis and lipolysis also contributed to defects in adipocyte differentiation.

Shockwaves induced preadipocytes to release extracellular ATP (Figure 5). Intracellular ATP is converted into cAMP through adenyl cylcases. cAMP is generally necessary for adipocyte differentiation along with IGF-1 and glucocorticoid [4]. cAMP is related to the production of endogenous PPARγ ligands [59] and regulates PPARγ transcriptional activity [60]. Increased cAMP activates cAMP-responsive element-binding protein (CREB), a transcription factor and expression of C/EBPα, which triggers expression of C/EBPα and PPARγ [61,62]. We found that cAMP levels declined in 3T3L-1 preadipocytes after shockwave treatment accompanied with extracellular ATP release (Figure 5). One hypothesis is that ATP release by shockwaves depletes intracellular ATP, resulting in decreased cAMP. Intracellular cAMP levels declined within 10 min after shockwave treatment. Shockwave treatment reduced the effect of IBMX on intracellular cAMP. Our results showed that shockwaves inhibited adipocyte differentiation through cAMP decrease during an early phase of differentiation (Figure 3 and Figure 5).

Differentiation of preadipocytes is regulated by a balance among local and endocrine factors that stimulate or inhibit differentiation [63]. Glucocorticoid agonists, insulin-stimulating IGF-1, PPARγ agonists, and cAMP are positive inducers of differentiation. In contrast, Wnt10b, TNFα, and TGFβ are negative regulators. Elevated cAMP concentration leads to differentiation through suppression of Wnt10b and specificity protein 1 (Sp1) and induction of C/EBPα [10,64]. Wnt10b is an endogenous activator of canonical Wnt signaling and Wnt10b in preadipocytes declines upon induction of differentiation [10]. In our in vitro system, extracellular ATP did not affect PPARγ expression (Figure 6) and thus shockwave-induced inhibition of adipogenesis appears to be caused by intracellular cAMP decrease. Overall, we demonstrated that shockwaves decreased cAMPs and led to conservation of Wnt10b and β-catenin, which suppressed PPARγ expression (Figure 7).

A Wnt/β-catenin pathway has been suggested as a negative regulator because of its role in inhibiting adipogenesis. Ectopic expression of Wnt10b inhibited adipocyte differentiation by suppressing the transcriptional factors, C/EBPα, and β-catenin [8]. β-catenin is the main regulating factor in the canonical Wnt pathway [32]. The activated Wnt/β-catenin pathway leads to an increase of β-catenin and a decrease of C/EBPα and PPARγ, the main transcriptional factors in adipogenesis. Therefore, Wnt signaling has been suggested as an attractive target for anti-obesity drugs [65]. Inhibition of PPARγ by an antagonist prevented high fat diet (HFD)-induced obesity and improved glucose and lipid metabolism [66]. Dominant-negative PPARγ mutation in humans and PPARγ deficiency in mice led to lipodystropy, and PPARγ deficient embryonic stem cells were unable to differentiate into adipocytes [67].

We evaluated the effects of shockwaves in adipogenesis and found that shockwaves suppressed differentiation of preadipocytes into adipocytes. Our study suggests low-energy shockwave treatment as a way to modulate a Wnt10b/β-catenin pathway and PPARγ expression, and provides new insights for the treatment of obesity.

## Figures and Tables

**Figure 1 cells-09-00166-f001:**
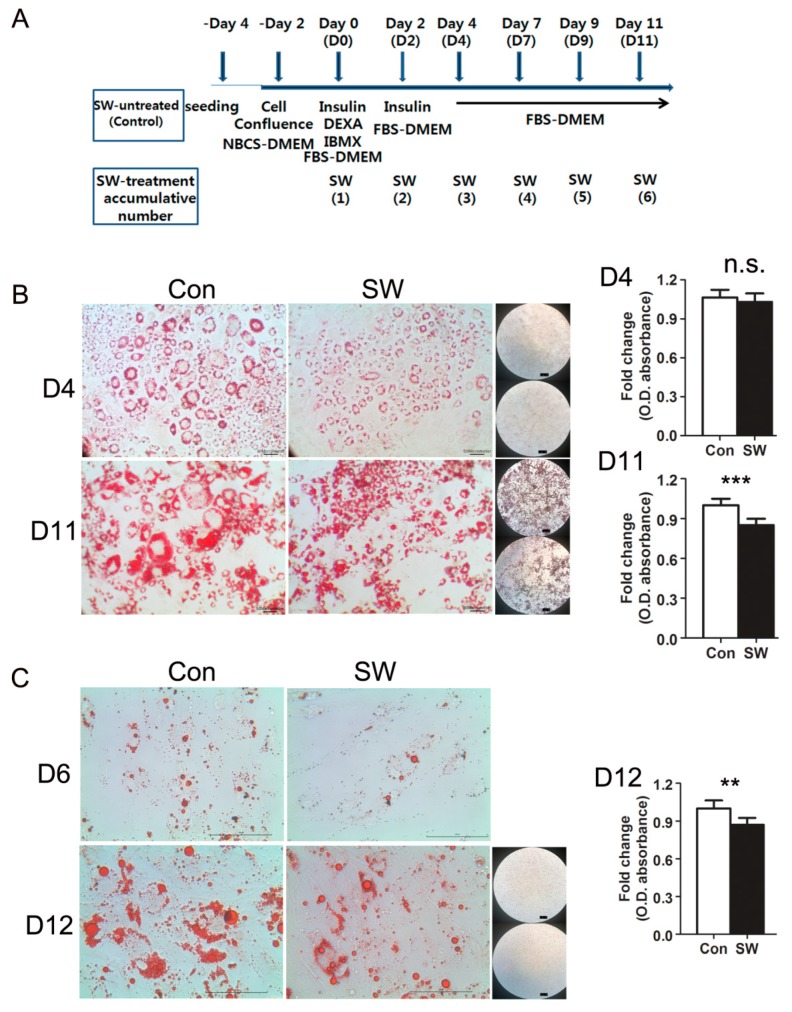
Shockwave treatment suppressed preadipocyte differentiation into adipocytes. (**A**) time chart of experimental procedure for differentiation. Two days after seeding, preadipocytes reached confluency. After two days, confluent preadipocytes were treated with or without shockwaves. Day 0 is the day of initiating differentiation and that the first dose of shockwave is given (D0). Shockwave treatment was done along with media changes (D2, D4, D7, D9, D11). Cells were harvested at the indicated timepoints. (**B**) 3T3L-1 cells were fixed and stained with Oil-red-O on days 4 and 11. Con; shockwave-untreated cells, SW; shockwave-treated cells. Representative photos are shown: 20×; scale bar, 50 μm (left two photos) 5×; Right black background, scale bar, 100 μm, upper; Con, lower; SW (right low-powered microscopic image). Quantification of Oil-red-O (O.D. of the isopropanol extraction), right graph. O.D. was normalized to DNA quantity; (**C**) Primary human subcutaneous preadipocytes were treated with or without shockwaves. Cells were stained with Oil-red-O on days 6 and 12. Representative photos are shown: 40×; scale bar, 100 μm (left two photos) 5×; Right black background, scale bar, 100 μm, upper; Con, lower; SW (right low-powered microscopic image). Quantification of Oil-red-O (O.D. of the extraction of isopropanol), right graph. O.D. was normalized by DNA quantity. Results were from triplicated experiments. In all graphs, scale bars are mean + S.D., ** *p* < 0.01, *** *p* < 0.001 (Student’s *t*-test). no significant; n.s., SW; shockwaves-treated cells, Con; untreated cells.

**Figure 2 cells-09-00166-f002:**
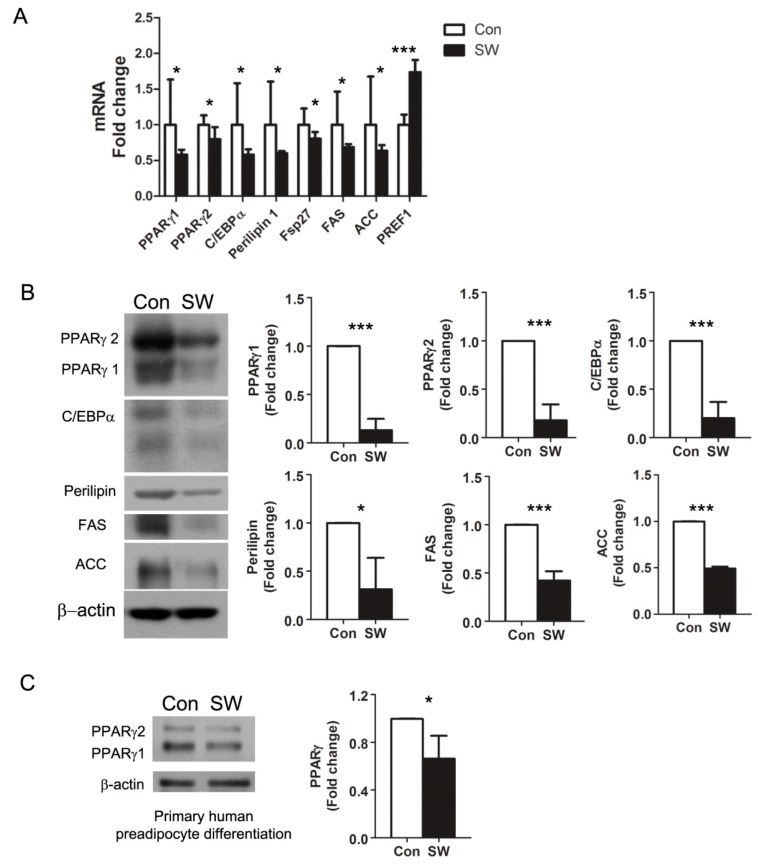
Shockwaves suppressed expression of adipocyte markers during preadipocyte differentiation into adipocytes. (**A**) quantitative RT-PCR. 3T3L-1 preadipocytes were treated with (SW) or without (Con) shockwaves as in Figure 1A. On day 12, RNA was extracted and cDNA was synthesized for qRT-PCR. PPARγ, C/EBPα, FAS, ACC, and PREF1; preadipocyte factor-1. (**B**) Western blots for adipogenesis markers were performed with 3T3L-1 cell lysates on day 12. (**C**) Primary human subcutaneous preadipocytes were treated (SW) or untreated (Con) with shockwaves as in Figure 1A. Immunoblotting for PPARγ was done on day 13. Right panels (**B**,**C**); quantification of band intensities. Results were from triplicated experiments. Fold-changes were normalized to β-actin or GAPDH. In all graphs, scale bars are mean + S.D., * *p* < 0.05, *** *p* < 0.001 (Student’s *t*-test).

**Figure 3 cells-09-00166-f003:**
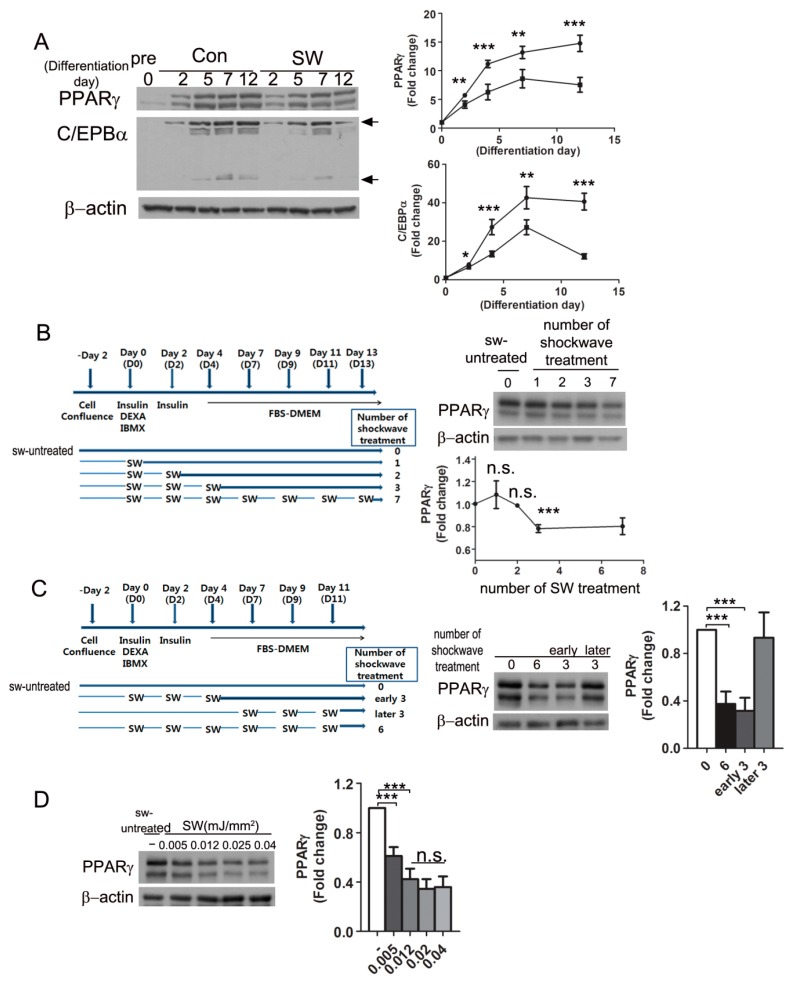
Shockwaves given at the initial stage of differentiation suppressed PPARγ expression. (**A**) shockwave-treated (SW) or untreated (Con) 3T3L-1 cells were harvested at indicated timepoints during differentiation and applied to Western blots. Right panel, time-course chagnes of PPARγ and C/EBPα. Shockwave treatment was done as in Figure 1A; (**B**) 3T3L-1 cells were treated with or without shockwaves as depicted in the chart. Cells were harvested on day 14 to PPARγ Western blots; lower graph, fold changes of PPARγ. The PPARγ level of shockwave-untreated cells (number of SW treatment; 0) were used for a baseline level of PPARγ (fold change 1.0); (**C**) Shockwaves were given as depicted in the chart. 3T3L-1 cells were harvested on day 12 and applied to PPARγ Western blots. The right graph is quantification of band intensities of PPARγ; (**D**) Shockwaves at different energy levels were given to 3T3L-1 preadipoctyes following the chart in Figure 1A. Cells were harvested on day 10 and applied to Western blots for PPARγ. Right panels are quantification of band intensities. In all graphs, scale bars are mean + S.D., * *p* < 0.05, ** *p* < 0.01, *** *p* < 0.001 (Student’s *t*-test). no significant; n.s.

**Figure 4 cells-09-00166-f004:**
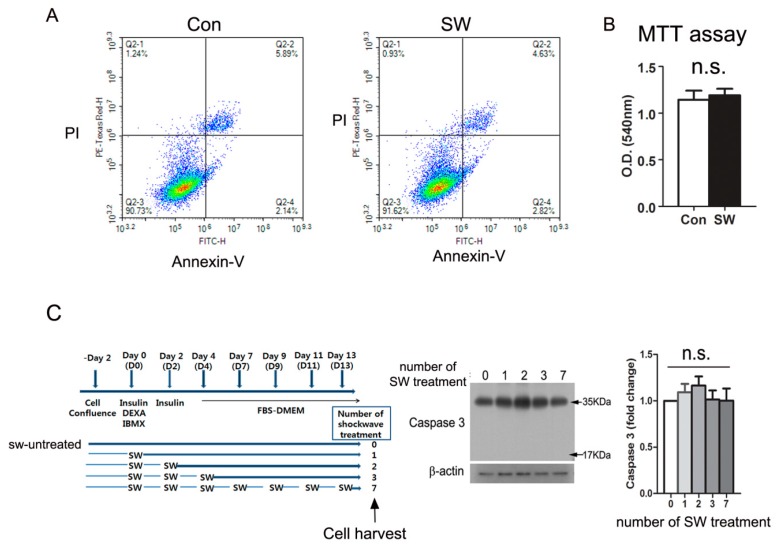
Low-energy shockwaves did not cause cell death. (**A**) 3T3L-1 preadipocytes were treated with or without shockwaves. The next day, cells were trypsinized and stained with Annexin V (FITC) and PI (Propidium Iodide). Flow cytometry was performed by using FACS Aria (BD Biosciences); (**B**) 3T3-L1 cells treated with or without shockwaves were used for the MTT assay on day 12. Results were represented as O.D. at 540 nm; (**C**) 3T3-L1 cells treated with or without shockwaves as depicted in the chart were harvested on day 14 and applied to immunblotting for caspase 3. β-actin was used for the internal control. Right graph; Quantification of the band intensities. Scale bars are mean + S.D., no significant; n.s.

**Figure 5 cells-09-00166-f005:**
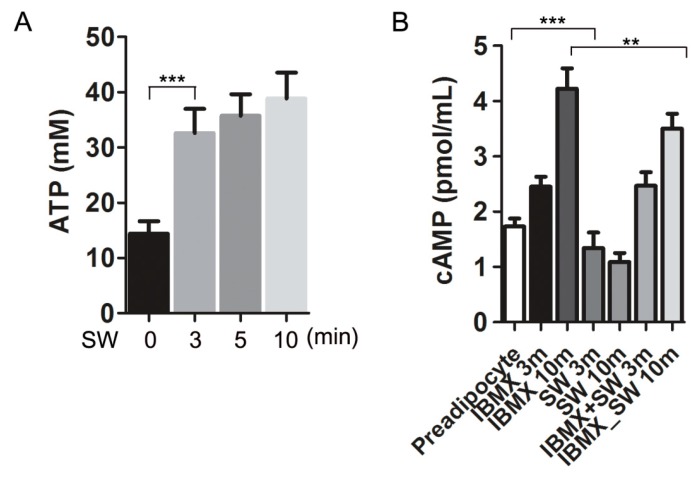
Shockwaves induced extracellular ATP release and decreased intracellular cAMP in 3T3L-1 preadipocytes. (**A**) 3T3L-1 preadipocytes were treated with or without shockwaves. After time points were indicated, extracellular ATP in the media was measured. (**B**) 3T3L-1 preadipocytes were treated with or without IBMX and/or shockwaves and harvested after 3 or 10 min. Intracellular cAMP levels were measured. Results are shown as mean + S.D. ** *p* < 0.01, *** *p* < 0.001 (Student’s *t*-test).

**Figure 6 cells-09-00166-f006:**
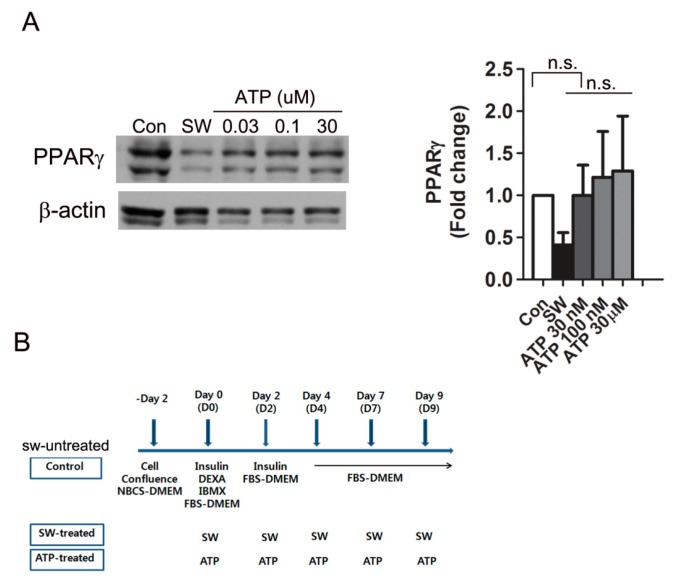
Extracellular ATP did not affect PPARγ expression. (**A**) Various concentrations of extracellular ATPs were added to 3T3-L1 cells instead of shockwave treatment. After 10 days of differentiation, cells were harvested and Western blots performed for PPARγ. Blots were quantified with the values normalized to PPARγ levels of untreated cells (Con). Results are shown as mean + S.D. in the graph; (**B**) chart for shockwave and ATP treatments in (**A**). no significant; n.s.

**Figure 7 cells-09-00166-f007:**
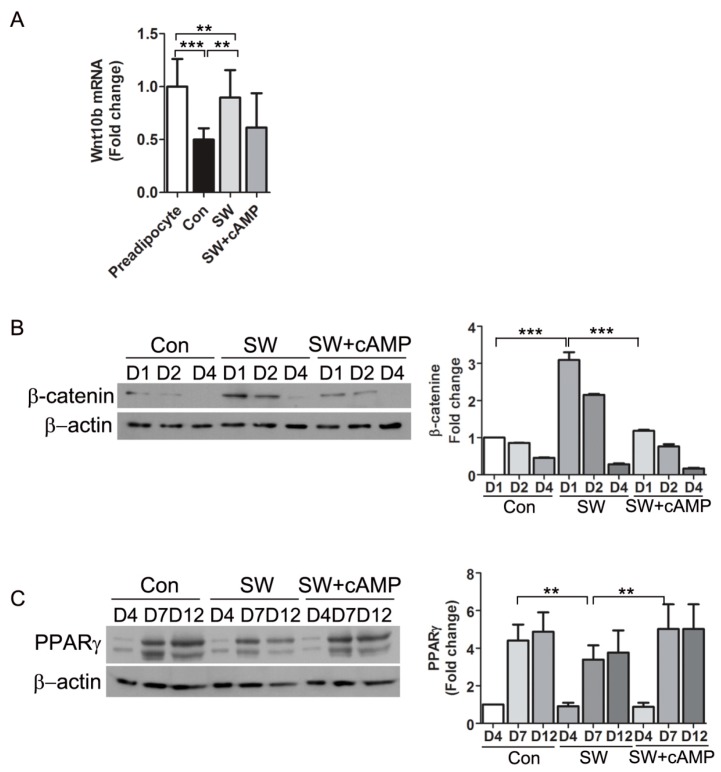
Shockwave-induced decrease of cAMP led to conservation of β-catenin and Wnt10b. (**A**) Shockwave-treated (SW) or untreated (Con) 3T3L-1 cells were harvested 6 h after induction of differentiation. The cAMP analog 8-bromo-cAMP (500 μM) was added into the medium right after shockwave treatment (SW + cAMP). Wnt10b RNA was measured by qRT-PCR. Preadipocytes without differentiation media were used for baseline Wnt10b level (preadipocyte); (**B**,**C**) shockwave-treated (SW) or untreated (Con) 3T3L-1 cells were harvested on days 1, 2, 4, 7 and 12. Cell lysates were prepared for immunoblotting for β-catenin and PPARγ. β-actin was the internal control. Right panels, quantifications of band intensities. Results are shown as mean + S.D. in all graphs. ** *p* < 0.01, *** *p* < 0.001 (Student’s *t*-test).

**Table 1 cells-09-00166-t001:** Primer list used in qRT-PCR.

	Forward	Reverse
PPARγ1	CAA GAA TAC CAA AGT GCG ATC AA	GAG CTG GGT CTT TTC AGA ATA ATA AG
PPARγ2	CCA GAG CAT GGT GCC TTC GCT	CAG CAA CCA TTG GGT CAG CTC
C/EBPα	AGC, AAC GAG TAC CGG GTA CG	TGT TTG GCT TTA TCT CGG CTC
perilipin	TGC TGG ATG GAG ACC TC	ACC GGC TCC ATG CTC CA
Fsp27	CTG GAG GAA GAT GGC ACA ATC GTG	CAG CCA ATA AAG TCC TGA GGG TTC A
FAS	GCT GCT GTT GGA AGT CAG C	AGT GTT CGT TCC TCG GAG TG
PREF1	CGG GAA ATT CTG CGA AAT AG	TGT GCA GGA GCA TTC GAT CT
ACC	GGA CAG ACT GAT CGC AGA GAA AG	TGG AGA GCC CCA CAC AC
Wnt10b	ACGACATGGACTTCGGAGAGAAGT	CATTCTCGCCTGGATGTCCC
GAPDH	AAC TTT GGC ATT GTG GAA GG	GGA TGC AGG GAT GAT GTT CT

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
