# Peer review of "Shockwaves Suppress Adipocyte Differentiation via Decrease in PPARγ"

_cells, 2020, doi:10.3390/cells9010166_

Round 1
Reviewer 1 Report
The manuscript “Shockwaves suppress adipocyte differentiation via decrease in PPARγ“ aims to evaluate the effects of shockwaves on adipogenesis by treating 3T3L-1 cells and human primary preadipocytes with shockwaves and analyzing adipocyte markers (PPARγ and C/EBPα) and extracellular ATP and intracellular cAMP levels. This study might give helpful ideas for the treatment of obesity with shockwaves.
Introduction:
- It would be valuable to cite more recent literature.
- Many reviews are cited. The authors should find literature that is more specific where appropriate.
- Some citations are missing, e.g. line 49 or line 60
- Regarding the beneficial effects of shockwave treatment only orthopedic diseases are mentioned and cited, more shockwave induced effects would be preferable.
-
Material & Methods:
- This section have to be strongly improved.
- The methods are very inaccurate, e.g. the paragraph “3T3L-1 cell culture and differentiation”. It is not clear which media is used for which condition, what was the cell density for differentiation, why are the human preadipocytes included here (rephrase title!). Human preadipocytes are never mentioned again in the methods section. Information on companies, concentrations and assays/kits (protein, cAMP measurement) is missing.
- The authors should choose a consistent description, e.g. once only “cells” are mentioned, then “cells treated with or without shockwaves”.
- It would be more clear to summarize the primers used in a table.
- The authors should think about to show SD instead of SEM.
Results/Figures:
- It is confusing why the cells were harvested at different time points (e.g. day 4 and day 11, day 6 and day 12).
- The quantitative analyses of Western blots should be verified or more representative images should be chosen as sometimes image and graph seem not to correlate.
- It is not clear why sometimes the control condition with untreated cells is not shown, e.g. Figure 3C.
- The authors should use uniform labeling, e.g. “control and SW-treated” and “Con and SW”.
- Sometimes the labeling is missing, e.g. Figure 3A-C.
- Figure legend 4: The information about Dexa and insulin in the medium should be included in the main manuscript, not (only) in the figure legend.
- Figure legends should be improved to become more uniform, e.g. once “shockwave-treated (SW) and 3T3L-1 preadipocytes (control)” are mentioned, then “shockwave-treated or untreated 3T3L-1 preadipocytes”.
- Figure S1A: Difference between solid line and dotted line is not visible. Maybe it is possible to include the image in higher quality or using different colors.
Discussion:
- It would be valuable to cite more recent literature.
- I would suggest to move some parts of this section into the introduction, e.g. line 310-312 or line 317-320.
- Figure S2 should not only be mentioned in the discussion section.
- the paragraph about mechanical forces (line 368-373) can be included in the first part of the discussion, as shockwave treatment is the major part of this study.
Reviewer 2 Report
The paper subject is interesting and authors suggest that shockwaves might modulate adipogenesis trough decreased PPAR and CEBP expression, decreased lipid droplet accumulation and inducing ATP releasing. In addition authors suggest that shockwaves tretament might be associated to obesity tretament.
However there are several point that are not well explained, or that are missed.
Concerning the introduction, it is very short and some information about cell mechanisms are missed. The hypothesis is not clear, the authors do not explain way using shockwaves might be an advance or not in obesity process. For example, in lines 58-61 authors said that the better understanding of adipogenesis mechanism shoud be important for obesity treatment, but they do not say why. The authors induces the reader that obesity might be controlled with decreased adipogenesis, however, the adipogenesis complex process and is important to accumulate lipids and to reduce the rates of free fat acid that might induce esteatosis or artherosclerose, among other effects. In this way, what is, if fact, the real proposal of the authors? How to decrease the adipocyte differentiation might help in top treat obesity condition? This information should be included in introduction and should be depper discussed in the discussion.
In lines 62-64 authors say that low energy shockwave therapy is benefical for some patologies but they did not do any details of why. What are the pathways modulated by shockwaves? What are the mechanisms that might be modulated or that are modulates, for example, in orthopedy? How exactly this kind of treatment works in cell context? After answer these questions, authors should explain why these therapy should be associated to obesity treatment, are the mecanisms modulated in orthopedy, for example, the same that should be modulated in obesity?
Concerning methodology section, it is very short and did not do details about cell maintainance, or other experiments.
For example, item 2.2 did not explain the conditions for differentiation and maintainance of human pre adipocytes. Authors said that they buy the cell from ATCC but did not give the catalog number of these cells, etc...
Item 2.4 said that author normalized the absorbance values of Oil red O (ORO) by protein concentration, which make no sense, if they are differentiating cells and proposing that cells might be in different stages, because protein content might be variable. They should present the results as absorbance and, if necessary, normalize these results by cell quantity (number of nucleus, or DNA quantity, or even by the absorbance of no differentiated cells- negative control, also missed in the experiment).
In item 2.6 authors did not indicate specifically the used antibodies.
Itens 2.7, 2.8 and 2.9 did not explain how exactly the experiments were performed, how the cells were maitained, what matherials and devices were used nor the particularities of the measurements.
Also in this topic, authors should explain the treatment strategy for each cells.
In results, all the figures are small and difficult to observe the result. There are experimental details in the legends that are not in the text.
In the first result (Figure 1), the measurement of ORO is wrong, should not be normilized by protein content. it is not clear how many independent experiments were performed and if they are performed in more than one replicate. It is impossible to read the legend of the Figure of the cells (1B and 1C) and figures seems to be in different scale, authors should put one image of the entire well next to the images in 20x. The scale bar in the figures are different from the one in the legend.
Why the 3T3 cells were collected in day 4 and 11? Why the authors did not collect the cells in the end of the differentiation (14th day)? Why the human preadipocyte were collected in day 6 and 12, when we know by several reports that the differentiation of this kind of cells take more than 21 days?
If these measurements were done they should be incorporated in the manuscript, if not, I strong recommend these measurements.
Concerning the treatment scheme, why the authors just treat cells in the begining of differentiation until the end? if shockwaves should make some difference in adipogenesis why did not to treta cells after they were differentiated? The authors should include this treatment condition to make sure that the SW might do a real effect in adipogenesis. If they consider that shockwaves might be an interesting treatment for obesity, the cells in this condition are differentiated and not in process of differentiation. in this way, to treat the cells after differentiation makes more sense.
Concerning the size measurement of the cells, there are no mention of how these measurementes were done. Authors should include this in the methodology. In addition, authors did not label nucleus and cytoplasm or membranes to make sure that the cells were well delimited, been very difficult to belive that they really measured the cells in a proper way. In this case, this measuremet should take in account the number of the cells and the label of the cells. In the way that this result is presented it should be discarded.
In addition, until this point authors did not show if SW could or not kill cells, the unique result concerning this fundamental question is very bad presented (Figure S1A), and also there is no mention about how this experiment was performed. The legend of the figure did not explain the experiment and there is no mentio to what is M1 and M2 or what is each curve in the plot. Also, anexin just show apoptosis and not show cell necrosis. It would be better if authors present a dot plot of anexin x propidium iodide to show if SW provokes apoptosis or necrosis. Also I think that this result is fundamental and should be in the main body of the manuscript, if the experiment will be done in a correct way.
In Figure 2, how the blots were performed? they were multiplex or the membrane was striped for each antibody? The entire membrane should be presented.
Concerning figure 3 the same problems are observed in the blots, also there are different colour in the membranes of the control and of the PPAR. The plots are bad represented in the legends and PPAr quantification seems not reproduce the blots. Also there are not error bars, does this means that this experiment was performed just one time without replicates? Legende must be redone.
Also, concerning Figure 3 issue, "Shockwave treatment at initial stage of adipocyte differentiation suppressed PPAR expression", why to treat adipocyte in the begining of the differentiation? Authors should include the treatment in the final stages or after differentiation.
What is the number of cells in the end of the differentiation? did the SW provoke detach cells?
Concerning Figure 4, did you measure the effect of SW in a no differentiated cell? Could the SW provoke the effect of ATP release in any kind of cells? did you measured this effect after a longer time (not only 3, 5 and 10 minutes) to observe if it is an short answer after SW and after hours the cells changes the ATP levels? How these changes in shorter times might provoke effects in differentiation if it takes 14 days?
In addition all the methodology, results section and legends should be improved giving more details of each experiment.
Finally, concerning the discussion, it is shallow, did not point fos pathway that are involved in the process nor the importance of the results. beyond the issues pointed in the begining of this review, lack of explanation of mechanism of SW in cells, why this should be important for obesity, how exactly it works, and why the therapies that use this tecnique were successful in terms of cell mecanisms, authors should link this success to obesity treatment os understanding. In sumary, explain why to use SW to understand or treat the obesity? Moreover, they should discuss why decrease adipogenesis and if it is really a good thing. Does the authors think that to inhibit adipogenesis should inhibit obesity?
The hypotesis should be improved and also the discussion.
Round 2
Reviewer 2 Report
The paper was improved.